# Transcriptome Analysis to Identify Genes Related to Flowering Reversion in Tomato

**DOI:** 10.3390/ijms23168992

**Published:** 2022-08-12

**Authors:** Yaoguang Sun, Wenhui Yang, Jinxiu Chen, Dexia Chen, Huanhuan Yang, Xiangyang Xu

**Affiliations:** Laboratory of Genetic Breeding in Tomato, College of Horticulture and Landscape Architecture, Northeast Agricultural University, Harbin 150030, China

**Keywords:** floral organ development, gene expression, RNA-seq, *Solanum lycopersicum*, vegetative and reproductive growth

## Abstract

Flowering reversion is a common phenomenon in plant development in which differentiated floral organs switch from reproductive growth to vegetative growth and ultimately form abnormal floral organs or vegetative organs. This greatly reduces tomato yield and quality. Research on this phenomenon has recently increased, but there is a lack of research at the molecular and gene expression levels. Here, transcriptomic analyses of the inflorescence meristem were performed in two kinds of materials at different developmental stages, and a total of 3223 differentially expressed genes (DEGs) were screened according to the different developmental stages and trajectories of the two materials. The analysis of database annotations showed that these DEGs were closely related to starch and sucrose metabolism, DNA replication and modification, plant hormone synthesis and signal transduction. It was further speculated that tomato flowering reversion may be related to various biological processes, such as cell signal transduction, energy metabolism and protein post-transcriptional regulation. Combined with the results of previous studies, our work showed that the gene expression levels of *CLE9*, *FA*, *PUCHI*, *UF*, *CLV3*, *LOB30*, *SFT*, *S-WOX9* and *SVP* were significantly different in the two materials. Endogenous hormone analysis and exogenous hormone treatment revealed a variety of plant hormones involved in flowering reversion in tomato. Thus, tomato flowering reversion was studied comprehensively by transcriptome analysis for the first time, providing new insights for the study of flower development regulation in tomato and other plants.

## 1. Introduction

In plants, the sexual reproduction process of flowering is a crucial developmental phenomenon in the life cycle [1,2]. It marks the transition of plants from vegetative to reproductive growth, a process commonly referred to as the flowering transition [3,4]. Initially, it was thought that flowering was an irreversible process in which the plant initiates a series of life activities, such as flower organ formation, pollination and fruiting, when it enters the reproductive growth stage [5]. Through ongoing scientific research, it has been discovered that in the presence of a variety of environmental factors (light, temperature, hormones, etc.) and genetic mutations, some plants enter the reproductive growth stage but then switch to vegetative growth development, a phenomenon known as flowering reversion [6,7,8]. The phenomenon of flowering reversion has attracted great interest and has been studied in tomato (*Solanum lycopersicum* L.), *Arabidopsis*
*thaliana* (L.) Heynh., rice (*Oryza sativa* L.), maize (*Zea mays* L.), soybean (*Glycine max* (L.) Merr.) and other crops [9,10,11,12,13]. Additionally, studies on related genes and gene regulation have gradually been implemented and refined in recent years [14,15].

The development of plant flowers involves the regulation of the balance between vegetative and reproductive growth, and the study of flowering reversion has been of great interest to scientists [16]. Thus far, the pathways affecting the induction of floral organ initiation can be divided into six types, related to photoperiod, vernalization, temperature, hormones, autonomy and age [17]. In the photoperiodic pathway, tomato is an intermediate day length plant and thus differs from the model plant *A. thaliana* in the process of flowering reversion. *SINGLE FLOWER TRUESS* (*SFT*) is a homolog of *FLOWERING LOCUS T* (*FT)* from *A. thaliana*. The *sft* mutation produces a late-flowering phenotype under both long-day and short-day conditions and causes the replacement of flowers by vegetative shoots [18]. *SELF PRUNING* (*SP*) is a gene homolog of *CENTRORADIALIS* (*CEN*)/*TERMINAL FLOWER1* (*TFL1*). The overexpression of *SP* in tomato causes flowers to be replaced by leaves in inflorescences and inhibits the transition from vegetative to reproductive growth [19,20]. In the vernalization pathway, the direction of meristem development in *Arabis paniculata* mustard is associated with the expression of floral organ characterization genes and the flowering initiation repressor gene *PERPETUAL FLOWERING 1* (*PEP1*) [21]. In longan (*Dimocarpus*
*longan* Lour.), flower development ceases under hot and humid winter conditions, and some inflorescences are transformed into nutritional branches [13,22]. Among the established hormonal pathways, gibberellin plays a role in promoting the initiation of flowering in plants, while cytokinins, auxins, methyl jasmonate and brassinolide play important roles in the development of flowering organs [23,24,25,26,27]. Studies on age pathways have mainly focused on perennials, whereas in annual or biennial plants more attention has been paid to the effects of environmental changes and gene regulation on flowering reversion [28].

The genes affecting flowering reversion can also be divided into two categories: flowering initiation-related genes and floral organ decision-related genes [29]. In recent years, many results have been obtained through the screening and study of floral organ development mutants [1,30]. The computational ordering of hundreds of tomato samples has allowed the floral transition process to be reconstructed at a fine temporal resolution and revealed that *uf* mutants exhibit a significant flowering reversion phenomenon. Unlike the *sft* and *dst* mutations, the depletion of *Uniflora* (*UF)* has a minor impact on flowering time but a major effect on shoot apical meristem (SAM) morphology during floral transition [31]. The downregulation of the expression of the SEPALLATA homolog *TM29* in tomato causes sepallata-like flowers, parthenocarpic fruit and ectopic shoots [32]. The *jointless* (*J*) mutation in tomato causes the inflorescence to form one to three flowers and then switch back to vegetative growth [33]. The failed termination of the floral meristem and the occurrence of flowering reversion in the *SlGT11* mutant indicate that *SlGT11* controls floral organ patterning and floral determinacy in tomato [34]. In *A. thaliana*, the *FT* and LEAFY (*LFY*) genes are involved in the photoperiodic regulation of flowering initiation [35]. The *ft* mutants undergo flowering reversion under short-day conditions, and *LF**Y* plays a similar role as a downstream gene of *FT* [36]. The *SHORT VEGETATIVE PHASE* (*SVP*) gene is a flowering repressor gene that causes flowering reversion when overexpressed in *A. thaliana* and *Petunia hybrida* [37]. In rice, mutations in the *OsMADS1* and *OsMADS15* genes cause the typical flowering reversion phenomenon [11]. In the maize indeterminate 1 (*id1*) mutant, inflorescence meristems undergo reversion to produce regenerative shoots, which may be associated with the accumulation of sugars [38]. In soybean, *GmBRI1*, *GmCPDs* and *GmNMH7* are associated with flowering reversion [39,40,41]. In switchgrass (*Panicum virgatum* L.), the simultaneous downregulation of *SPL7* and *SPL8* expression results in flowering reversion [42]. These studies of mutants and gene function have contributed to our understanding of flowering reversion, yet comprehensive knowledge of gene expression patterns during flowering reversion is still lacking in tomato.

Transcriptome sequencing is an effective technique for examining complex biological pathways and molecular mechanisms of gene expression networks by systematically studying gene transcription at the entire transcriptional level [43]. In this study, we selected the ‘116’ and ‘117’ cultivars as research materials, among which the ‘117’ variety is highly susceptible to flowering reversion under the same growing conditions as ‘116’. The sampling time points were set at the flowering induction period, flower bud differentiation period and floral organ formation and developmental period, and inflorescence meristems were collected. The RNA sequencing (RNA-seq) platform was used to analyze differentially expressed genes (DEGs) and to mine and screen candidate genes associated with flowering reversion. Finally, the changes in the contents of endogenous hormones were measured, and the plants were treated with exogenous hormones. Our primary aim was to gain an overall understanding of the genes that regulate flowering reversion, which could benefit tomato breeding for producing optimal strains in the future.

## 2. Results

### 2.1. Phenotypic Comparison between ‘116’ and ‘117’

Tomato flowering reversion occurs at the intersection at the top of the inflorescence and produces branches or leaves, which gradually grow along with the flowers and fruits as time progresses. As shown in Figure 1, flowering reversion was observed in ‘117’ plants (Figure 1A), whereas no obvious signs were observed in ‘116’ plants (Figure 1B). Flowering reversion became evident during the inflorescence period, and nutrient branches started to grow in the fruiting and expansion stages. During the fruit-ripening period, there were very obvious nutritional branches at the anterior end of the topmost fruit. The changes in the inflorescence meristems of ‘116’ and ‘117’ were observed by light microscopy. As shown in Figure 1C–H, in ‘117’, the inflorescence axis was divided in two, and the portion in which flowering reversion occurred contained more cells with larger diameters. The diameter was more consistent in ‘116’ plants.

### 2.2. Quality Assessment and Repeat Correlation Analysis of RNA-seq Data

A total of 119.72 Gb of clean data were obtained after data filtering; each sample yielded an average of 6.65 Gb of data, and the percentage of Q30 bases in each sample was not less than 92.55%. The raw data composition of the raw reads obtained from sequencing is shown in Appendix A. The specific sequencing data quality of the specific sequenced samples is shown in Appendix A. Overall, all 18 samples showed good sequencing quality and fully met the requirements for subsequent data analysis. Additionally, the raw reads obtained from the RNA sequencing of all 18 samples were submitted to the NCBI Sequence Read Archive database with the registration number SUB10254346.

After sequencing and comparison with the reference genome (the tomato genome version SL 4.0 and annotation ITAG 4.0), a total of six groups were finally screened in different time periods, and the numbers of genes identified are shown in Appendix A. The lowest number of known genes accounted for 71.07%, and the total number of all genes in each group was greater than 24,000. Pearson correlation analysis was performed on 18 samples of the two studied varieties, ‘116’ and ‘117’, in three periods; from the results (Figure 2), which show a correlation coefficient of 0.6 < R^2^ < 1, we concluded that the 18 samples presented a relatively high degree of homogeneity of genes within each sample. R^2^ < 0.6 indicates poor reproducibility. Although several samples in the analysis showed R^2^ < 0.6, this did not affect our subsequent bioinformatic analysis. From the reproducibility tests, we learned that the correlation between the reproducibility of the two cultivars was strongly influenced by time variation.

### 2.3. Analysis of DEGs in Different Comparison Groups

The intra- (Early_116 vs. Mid_116, Early_116 vs. Late_116, Mid_116 vs. Late_116; Early_117 vs. Mid_117, Early_117 vs. Late_117, Mid_117 vs. Late_117) and intergroup (Early_116 vs. Early_117, Mid_116 vs. Mid_117, Late_116 vs. Late_116) comparisons of DEGs between the 18 samples of ‘116’ (no flowering reversion) and ‘117’ (flowering reversion) in different periods provide a clearer understanding of the up- and downregulation patterns of DEGs in the different comparative analyses, as detailed in Appendix A and Figure 3A.

In ‘116’, the interperiod comparisons revealed 2, 2047 and 2067 upregulated genes and 3, 741 and 852 downregulated genes in the early, middle and late stages, respectively, while in ‘117’ the same comparison revealed 22, 3 and 18 upregulated genes and 4, 3 and 45 downregulated genes (Figure 3A). The number of DEGs in ‘116’ was significantly higher than that in ‘117’ and much higher in the middle and late stages than in the early stages, indicating that a large number of DEGs were involved in the flowering reversion pathway in the middle and late stages and that the late stage was the main stage of differential gene expression. Similar results were found in the late-stage intergroup comparisons, where a total of 31 DEGs were upregulated and 22 DEGs were downregulated in the mid-stage of the intergroup comparisons; in the late stage, the number of upregulated DEGs reached 839 and the number of downregulated DEGs reached 2313. The identified intergroup differences are shown in more detail in Figure 3B. The proximity indicated that the mid–late stage is the main period of flowering reversion, and this period is also the main stage of gene activation for meristem determination, floral organ characteristic determination and floral organ formation.

The VENN statistics of the comparative analysis between groups showed that the number of DEGs shared between the pre- and mid-phases was 31, while the number of DEGs shared with the post-phase was 57 and the number of DEGs shared between the mid- and post-phases was 29 (Figure 3C). This further indicated that the large numbers of DEGs associated with flowering reversion are expressed in the middle and late stages and that these genes are continuously and stably expressed during the maintenance of flowering, which may be the mechanism whereby plants maintain their flowering status. The specific expression levels and details of these DEGs are shown in Appendix A.

### 2.4. GO and KEGG Enrichment Analysis of DEGs

GO analysis was performed on the set of DEGs identified between the two cultivars at different times with a *p*-value ≤ 0.05 (Figure 4; Appendix A). The GO analysis results were divided into three main categories: biological process, cellular component and molecular function. In the classification of biological processes, DEGs were annotated to a total of 17 significant entries. The metabolic process, cellular process, localization, biological regulation, regulation of biological process and response to stimulus entries were enriched with large numbers of DEGs (≥150 DEGs), especially in the metabolic process category, which included the most enriched DEGs. In the classification of cellular components, DEGs were annotated to a total of 11 significant entries, among which the DEGs were abundantly enriched in the membrane, cell, cell part, organelle, protein-containing complex and membrane part entries. In the classification of molecular function, DEGs were annotated to 10 significant entries. The entries for binding, catalytic activity, transporter activity and structural molecule activity were associated with a large number of DEGs. Thus, the GO classification analysis suggested that DEGs associated with flowering reversion in tomato may be involved in cellular metabolism, cellular components, binding and catalysis.

KEGG pathway enrichment analysis was also performed on the set of DEGs screened via transcriptome sequencing in three different periods for both the ‘116’ and ‘117’ varieties (Figure 5, Appendix A). The KEGG classifications were divided into five categories: cellular processes, environmental information processing, genetic information processing, metabolism and organismal systems (Figure 5A). In the cellular processes category, transport and catabolism (256 DEGs) and cell growth and death (82 DEGs) were significantly enriched. In the environmental information processing category, signal transduction (284 DEGs) was significantly enriched. In the genetic information processing category, translation (352 DEGs) and folding as well as sorting and degradation (248 DEGs) entries were significantly enriched. The largest number of DEGs was enriched in the metabolism category, and these genes were concentrated in the carbohydrate metabolism (540 DEGs), carbohydrate metabolism (540 DEGs), global and overview maps (335 DEGs), amino acid metabolism (272 DEGs) and lipid metabolism (271 DEGs) entries. During the KEGG pathway enrichment analysis of these DEGs, the top 20 most significant pathways were selected by ranking the significantly enriched pathways from lowest to highest according to Q-values ≤ 0.05. The biosynthesis of amino acids (ko01230, 379 DEGs), starch and sucrose metabolism (ko00500, 288 DEGs), glycerophospholipid metabolism (ko00564, 142 DEGs), carbon metabolism (ko01200, 424 DEGs) and glycolysis/gluconeogenesis (ko00010, 212 DEGs) pathways were significantly enriched, as shown in Figure 5B.

To gain more insight into the enrichment of these DEGs in KEGG pathways, we performed separate KEGG enrichment analyses of the DEGs from the three different developmental stages of the two cultivars, as shown in Figure 6. In the early stage, these DEGs were mainly concentrated in the DNA replication, cell cycle, meiosis and citrate cycle (TCA cycle) categories (Figure 6A). In the middle stage, citrate cycle (TCA cycle) entries were still the most enriched entries, except for carbon metabolism and pentose and glucuronate interconversions, which were also enriched with large numbers of DEGs (Figure 6B). In the later period, the most enriched entries were ribosome, biosynthesis of amino acids, biosynthesis of antibiotics, carbon metabolism, starch and sucrose metabolism and photosynthesis antenna proteins (Figure 6C).

### 2.5. Identification of Key Regulatory Genes for Flower Formation Reversal in Tomato

A large number of genes have been shown to be involved in the regulation of flowering organ formation in tomato flowering reversion studies; based on previous studies, we summarized 351 flowering marker genes according to their expression in the transcriptome, as shown in Figure 7 and Appendix A [31]. In the KEGG analysis, we identified DEGs enriched in photosynthesis, cell growth and death, and aging pathways, which may be factors regulating flowering reversion in tomato; the details of these results are shown in Appendix A. Additionally, in the GO analysis, we identified three genes related to cell proliferation—*Solyc01g106830*, *Solyc02g092110* and *Solyc10g083580*—among the DEGs.

### 2.6. WGCNA of DEGs

To further investigate the interregulation of DEGs during flowering reversion in ‘116’ and ‘117’ plants at different times, a total of 3223 genes were screened for the construction of scale-free co-expression networks, and according to the optimal power threshold for module mining and co-expression topology, heatmap construction was performed according to the optimal power threshold (Figure 8A), and 18 co-expression modules were established (Figure 8B). Expression pattern analysis of genes within modules showed that genes within the greenyellow, grey60 and sienna3 modules were highly expressed in ‘116’ plants (Pearson correlation coefficient ≥0.8), and we performed KEGG enrichment analysis of the genes within these modules. The results are shown in Appendix A. Genes within the yellowgreen, brown, greenyellow, darkred and grey60 modules were highly expressed in the Late_116 group, suggesting that a relatively high expression trend for these genes may be correlated with flowering time; the darkgreen, pink, saddlebrown and darkmagenta modules were highly expressed in sample ‘117’, and genes within the pink module were highly expressed within the Late_117 group. The mining of these genes may provide a basis for explaining tomato flowering reversion.

### 2.7. Analysis of the Metabolic and Regulatory Pathways of DEGs

MapMan software was used to visualize the results of the enrichment analysis of DEGs between the ‘Late’ groups to explore their regulatory pathways (Figure 9 and Appendix A). As shown in Figure 9, most DEGs were upregulated, and these DEGs were mainly associated with such entries as the cell wall, lipid metabolism, carbohydrate metabolism and amino acid metabolism. In addition, some genes in the photosynthesis pathway were downregulated. Hormones such as IAA, abscisic acid (ABA), cytokinins and GA play an important role in the process of flower formation reversal. These hormones have important functions in plant flowering and flower organ formation. In the IAA hormone pathway, 6-BA-related genes were upregulated and GA-related genes were downregulated.

### 2.8. Validation of RNA-Seq Data by RT-qPCR

To verify the reliability of the RNA-seq results, we selected nine DEGs for RT-qPCR validation. These nine DEGs were mainly selected from 351 genes related to flowering, including cell differentiation-related genes, tomato flower formation regulatory genes, and tomato flowering repressor genes. As shown in Figure 10, the RT-qPCR and RNA-seq data showed similar trends, confirming the accuracy of the RNA-seq results. Among these nine DEGs, the *CLE9* and *CLV3* genes were expressed at low levels, which is consistent with their expression patterns as cell differentiation-related genes. The expression levels of all genes except *SVP* were higher in ‘116’ than in ‘117’, with *S-WOX9* showing the highest expression. The expression pattern of *SVP* differed from that of the other genes, as it showed higher expression levels in ‘117’ than in ‘116’.

### 2.9. Measurements of GA3, IAA and 6-BA Hormones

In the endogenous hormone analysis, we measured the contents of Gibberellin A3 (GA3), indole-3-acetic acid (IAA) and 6-Benzylaminopurine (6-BA). The test results showed that the expression levels of the three hormones were different between the two varieties (Figure 11). The content of GA3 showed a decreasing trend in ‘117’ and an increasing trend in ‘116’, and the content in the middle and late periods was higher in ‘116’. The levels of IAA and 6-BA showed similar trends; they were higher in ‘117’ than in ‘116’ and presented a decreasing trend in ‘117’ and an increasing trend in ‘116’.

### 2.10. Effects of Hormones on Flowering Reversion in Tomato

Tomato flowering reversion significantly changed the phenotypes of the two experimental materials treated with different concentrations of GA3, IAA and 6-BA (Figure 12). After GA3 treatment, plant height increased significantly and flowering reversion became more obvious, with a large number of vegetative branches appearing at the tips of the inflorescences; however, flowering was not reversed in ‘116’. After IAA treatment, the most obvious change was the reversal of flowering in ‘116’ plants. After 6-BA treatment, although flowering reversion was not observed, the vegetative branches grew rapidly. With increasing concentrations of the three hormones, the development of flowering reversion became more obvious and vegetative branches appeared earlier.

## 3. Discussion

### 3.1. Transcriptomic Analysis Yields 3223 DEGs Associated with Flowering Reversion

Flowering reversion is a phenomenon in which differentiated floral organs and tissues show a reversal of their development trajectories and enter vegetative growth again, which has been found in many species, including both monocotyledons and dicotyledons [8,44]. In the study of the molecular mechanism of floral organogenesis, scientists have found a variety of genes related to flowering reversion by investigating flowering reversion-related mutants and flower development [45,46,47]. However, flowering reversion is not simply the reversion of flower development but also may involve other novel biological regulatory processes. Therefore, it is necessary to conduct a more comprehensive molecular biological analysis of flowering reversion from a new point of view. In this study, transcriptome analysis was carried out on the inflorescence meristem, which readily underwent flowering reversion in ‘117’ but did not show flowering reversion in ‘116’. According to the developmental stages and different developmental characteristics of the different materials, it can be inferred that the DEGs identified in ‘116’/’117’ are related to flowering reversion. A total of 3223 DEGs that may be involved in tomato flowering reversion were screened, among which 25 genes showed differences in gene expression in the three periods.

Through GO enrichment, KRGG pathway enrichment and WGCNA of the 3223 DEGs screened, it was found that many biological processes, such as cell signal transduction, gene transcriptional regulation, protein post-transcriptional regulation and metabolic pathway regulation, were involved in the induction of tomato flowering reversion. The results showed that the induction of tomato flowering reversion is a complex phenomenon involving many biological processes. In addition, this study revealed that a large number of DEGs were enriched in classifications related to organelles and membrane structure, and a large number of DEGs were also enriched in plant signal transduction-related classifications. Thus, increased signal transduction could be observed among tomato inflorescence meristem cells undergoing flowering reversion.

### 3.2. The Relationship between Floral Organ Development and Flowering Reversion

Floral organ development refers to the process by which the stem tip or axillary meristem changes from vegetative growth to reproductive growth and finally develops into florets under the joint action of appropriate internal and external factors when the plant enters a certain growth period. This process involves the developmental transformation of the stem tip meristem and the transformation of spikelet meristem to spikelet meristem, spikelet meristem to floret meristem and floret meristem to primordial meristem in each whorl organ. Flowering reversion can occur at different stages of floral organ development when internal and external environmental conditions are not suitable for the further development of floral organs; in this process, the floret primordium, spikelet primordium or inflorescence primordium may partially or completely re-enter vegetative growth, rather than gradually degenerate to a higher level of tissue formation [48,49,50]. In addition to the reversion of these tissues located at the top of organs, other types of organs that are not fully differentiated (such as the bract primordium, calyx primordium and petal primordium) can also undergo reversal; they will eventually lose the ability to develop into regenerated buds, such as the apical tissues of organs, and will instead form leaves or leaf-like vegetative organs [51].

In this study, through the analysis of the expression of homologous genes related to flower development in tomato, it was found that the screened DEGs were consistent with 58 genes related to flower development. The relative expression of some key genes in floral organ development was analyzed, and it was found that the expression of *CLE9* and *CLV3*, which are related to cell differentiation, was lower in ‘117’ than in ‘116’. *FA* and *SFT*, which are homologous genes of the flowering-promoting genes *LFY* and *FT*, were expressed at very low levels in the two materials but were expressed at higher levels in ‘116’. The expression patterns of the *PUCHI*, *UF*, *LOB30* and *S-WOX9* genes were similar. In contrast to other genes, the flowering suppressor gene *SVP* was highly expressed in ‘117’. These results are similar to those of previous studies indicating that there are large differences between the molecular mechanisms of floral organ development and flowering reversion [31,52]. Previous studies have also found that the expression patterns of genes related to flower development in plants undergoing flowering reversion are not completely opposite those in normal flowering plants and that there is still some relationship between them [10,14,53].

### 3.3. Effect of Plant Hormones on Flowering Reversion

Plant hormones are small molecular compounds whose contents in plants are very low, but they are nevertheless involved in all aspects of plant development [54]. Gibberellin, cytokinins, auxins, methyl jasmonate and brassinolide all play important roles in the development of plant floral organs [55]. Plant hormones also play an important role in the process of flowering reversion; for example, in longan, the levels of cytokinins in materials undergoing flowering reversion are lower than those in materials not showing this phenomenon; in contrast, the contents of gibberellin and abscisic acid are higher [13]. The overexpression of cytokinin synthesis genes leads to the abnormal development of floral organs in *Arabidopsis thaliana* [24].

The regulation of plant development by different hormones is not completely independent, and there are a variety of complex relationships between these pathways. For example, the elimination of proteins needed for the functions of hormones, such as auxins, ethylene and jasmonic acid, is related to the activity of the ubiquitin 26S proteasome (ubiquitin-26S proteasome) [56]. In addition, different hormones may show competitive or antagonistic functions. Cytokinins and auxins play an indispensable role in plant development, and the ratio of CTK/AUX determines the developmental trajectories of the plant apical meristem, root apical meristem and tissue cultured in vitro [57]. Abscisic acid, gibberellin and ethylene play antagonistic roles in the production of phenolic acids in the hairy roots of *Salvia miltiorrhiza*. The application of one hormone alone can effectively induce an increase in phenolic acid content, but when two kinds of hormones are added at the same time the contents of different types of phenolic acids are significantly affected [58].

In this study, we determined the contents of endogenous hormones (GA3, IAA and 6-BA) in tomato inflorescence meristems. There were significant differences in the contents of the three hormones in the early stage, which were higher than those in ‘117’, indicating that they were involved in the process of flower formation reversion. In ‘116’ plants, the contents of the three hormones increased gradually; in contrast, the contents of 6-BA and IAA remained high in ‘117’ plants. The results of exogenous hormone induction treatment showed that there were significant changes in flowering reversion after treatment with the three hormones. GA3 and 6-BA may be related to growth after flowering reversion, and IAA may be involved in the process of flowering reversion. The results suggest that tomato flowering reversion may be affected by multiple hormones or controlled by multiple hormone-related genes.

### 3.4. Flowering Reversion Is the Result of Multiple Biological Processes

In the study of floral organ development, it has been established that transformation from vegetative growth to reproductive growth involves flowering signal perception, signal transmission, flowering initiation and floral organ determination-related genes, which determine the developmental direction of the meristem in different periods. In the study of flowering reversion in plants, several genes involved in flowering reversion have been found through the study of mutants or reverse genetics; for example, flowering reversion has been studied in in *Arabidopsis thaliana ft* and *lfy-6* mutants [59,60], flowering reversion has been studies in rice *OsMADS15* and *OsMADS1* mutants [61], and floret reversion in the inflorescence meristems of maize has been studied in *id1* mutants [62]. However, these flowering reversion phenomena can also be associated with flower development to some degree. In addition, flowering reversion may involve several biological processes, such as the inhibition of further floral organ development, regulation of differentiated floral meristem development and promotion of vegetative growth [63]. Although this study did not prove the existence of multiple pathways of flowering reversion under experimental conditions, on the basis of previous studies and our transcriptome analysis, it can be inferred that there are many biological pathways related to flowering reversion in tomato.

## 4. Materials and Methods

### 4.1. Plant Materials and Growth

The tomato cultivars ‘116’ and ‘117’ used in this study were obtained from our laboratory. The plants were grown in a solar greenhouse located at the experimental site of the Northeastern Agricultural University, Harbin, China (126.916 E, 45.773 N). Four-week-old seedlings were transplanted, and regular water and fertilizer management were then applied until flowering and fruiting. Randomized complete block designs were used, with 10 plants planted in each plot and three replications. The sampled tissue was selected from the inflorescence meristem of the second or third spikes of flowers, and sampling was performed three times, during flowering induction, floral bud differentiation and floral organ formation and development [31]. The collected tissues were immediately immersed in liquid nitrogen and subsequently stored at −80 °C until use.

### 4.2. Microscopic Structure Analysis of Tomato Flowering Reversion

On the basis of anatomical analysis, histological analyses were performed according to the different periods of the sampled inflorescence meristem. The inflorescence meristems were fixed in FAA (formalin:acetic acid:70% alcohol = 1:1:18) fixative for 8 h, followed by immersion in Ehrlich Hematoxylin solution for 3 days [41]. After the completion of staining, the tissues were rinsed with distilled water for 1.5 h, followed by decolorization in an ethanol gradient (50, 70, 80 and 90% for 20 min each, followed by 100% for 15 min two times) and clearing treatment with xylene (ethanol:xylene = 1:1 solution for 1.5 h, followed by 3 treatments with pure xylene for 1.5 h each), after which the samples were embedded, sectioned and imaged [10]. Imaging was performed under an Olympus BX53 light microscope (BX53, Olympus, Tokyo, Japan).

### 4.3. mRNA Library Construction and Sequencing

The extraction and detection of total RNA, the ethanol precipitation protocol and CTAB-pBIOZOL (product code: BSC55S1, Bioer Technology, Hangzhou, China) reagent were used for the purification of total RNA from the plant tissue, according to the manufacturer’s instructions. Subsequently, the total RNA was qualified and quantified using a NanoDrop instrument and Agilent 2100 Bioanalyzer (Thermo Fisher Scientific, Waltham, MA, USA) [64]. For the construction of mRNA libraries, oligo(dT)-attached magnetic beads (L-3002A) were used to purify mRNA (bio-Linkedin, Shanghai, China). The final library was amplified with phi29 (D7053L) to produce DNA nanoballs (DNBs) containing more than 300 copies of one molecule (Beyotiom, Shanghai, China) [65]. DNBs were loaded into the patterned nanoarray and paired-end 150-base reads were generated on the MGISEQ-2000 platform (BGI, Shenzhen, China) [66]. Eighteen libraries were constructed, representing six biological variants and three replicates.

### 4.4. Mapping Reads and DEG Analysis

The sequencing data were filtered with SOAPnuke (v1.5.2) by removing (1) reads containing sequencing adapters, (2) reads whose low-quality base ratio (base quality less than or equal to 5) was more than 20% and (3) reads whose unknown base (‘N’ base) ratio was more than 5%. Then, clean reads were obtained and stored in FASTQ format [67]. The clean reads were mapped to the *S. lycopersicum* reference genome sequence (tomato genome version SL 4.0 and annotation ITAG 4.0) using HISAT2 (v2.0.4) [68]. Bowtie2 (v2.2.5) was applied to align the clean reads to the reference coding gene set, and the expression levels of the genes were calculated by RSEM (v1.2.12) [69]. DEGs were identified using DESeq2 (v1.4.5) according to a Q-value ≤ 0.05 and |log2FC| ≥ 2 [70].

### 4.5. Gene Ontology (GO) Functional and KEGG Pathway Enrichment Analysis of DEGs

To gain insight into the observed phenotypic changes, GO (http://www.geneontology.org/) (accessed on 2 April 2022) and KEGG (https://www.kegg.jp/) (accessed on 2 April 2022) enrichment analyses of annotated DEGs were performed using Phyper (https://en.wikipedia.org/wiki/Hypergeometric_distribution) (accessed on 10 April 2022) based on the hypergeometric test [71]. The significance levels of terms and pathways were subjected to Bonferroni correction based on the Q-value with a rigorous threshold (Q-value ≤ 0.05).

### 4.6. Weighted Gene Co-Expression Network Analysis (WGCNA)

The WGCNA version 1.70–3 R package was used to perform WGCNA based on the expression correlation pattern between DEGs [72]. The analysis was applied to all DEGs, and log2 (FPKM + 1) values were used as the input. Soft thresholds were set as the optimal values (SET. R. sq value > 0.8 and slope value close to −1) to make the network suitable for a scale-free topology. The minimum number of genes within a module was 25 (minModuleSize = 25), and the similar module merging threshold was 0.25 (cutHeight = 0.25). Based on the Pearson correlation coefficient between samples, we chose Pearson values > 0.8 as the analysis module for further discussion.

### 4.7. Analysis of MapMan Biological Functions of DEGs

MapMan is software used for the functional analysis of plant genes and pathway analysis that enables the integration and visualization of the functions of DEGs in metabolic pathways [73]. We downloaded ITAG (v4.0) Mapping and Pathways from the MapMan Store and loaded them into the appropriate locations in the software. When using ITAG (v4.0) mapping for X4 annotation, only those pathways marked as X4 were used. The experimental matrix (i.e., FPKM values of DEGs in the transcriptome) was loaded, and the newly loaded DEG data were employed as the matrix for microarray analysis in MapMan. The experimental data were displayed along the pathway, after which the corresponding Pathways file and then X4.4 *Solanum lycopersicum* mapping were selected, all adjustments were completed and the image was outputted.

### 4.8. Expression Profiles of Plant Flowering-Related Marker Genes

By combining previous studies and our preliminary experimental results, 351 genes related to plant flowering were finally selected [31]. The FPKM expression values of these genes were then row-normalized and used to generate heatmaps.

### 4.9. Real-Time qPCR Analysis

Total RNA was extracted using CTAB-pBIOZOL reagent, following the methodological steps in the accompanying instructions. cDNA was synthesized using the Transcript II One-Step gDNA Removal and cDNA Synthesis Kit (TransGen Biotech, Beijing, China). In this study, all specific PCR primers were designed using Primer Premier 5 software, as shown in Appendix A. The *EF1α* gene was used as an internal reference gene [74]. The experiments were performed using AceQ^®^ qPCR SYBR^®^ Green Master Mix (Vazyme, Nanjing, China) and a qTOWER^3^G detection system (Analytik Jena, Thuringia, Germany). The PCR reaction was performed as follows: 10 min at 95 °C and 40 cycles of 94 °C for 20 s and 60 °C for 30 s. Expression analysis of the genes was performed using the 2^−∆∆CT^ method, and significance analysis was performed using SPSS 7.0.

### 4.10. Measurements of Relevant Hormone Contents

Inflorescence meristems were collected in each of the above three periods three times in each experimental group from both cultivars. The samples were extracted and purified according to the method of Li et al. [75]. GA3 (B20187), IAA (B21810) and 6-BA (B24213) standards from Sigma (Yuanye Bio-Technology Co., Ltd., Shanghai, China) were used in these analyses.

### 4.11. Exogenous Hormone Treatment

Tomato plants were sprayed with different concentrations of exogenous hormones at the seedling stage. The three hormones were GA3, IAA and 6-BA, consistent with the determination of endogenous hormones. The GA3 application concentrations were 100, 150 and 200 mg·L^−1^, and the IAA and 6-BA application concentrations were 50, 100 and 150 mg·L^−1^. Thirty plants with the same growth were selected for each hormone treatment, and they were divided into 3 groups with 10 plants in each group, three replications were performed and the whole experiment was repeated three times, with growth status and flowering reversion recorded every week.

## 5. Conclusions

In this study, through the phenotypic and microscopic observation of flowering reversion in tomato, it was found that, during flowering reversion, the inflorescence meristem formed vegetative branches. The transcriptomic analysis of inflorescence meristems in three different developmental stages of tomato materials ‘117’ (showing flowering reversion) and ‘116’ (showing normal development) resulted in the screening of 3223 DEGs. Through the multidatabase annotation of these DEGs, it was found that multiple biological processes, such as cell signaling, gene transcription regulation, protein post-transcriptional regulation and metabolic pathways, were involved in the reversion of tomato flowering. In addition, a large number of DEGs were enriched in organelle- and membrane-associated categories, and from this, combined with a large number of DEGs enriched in plant signaling-related categories, it can be speculated that these DEGs may play a role in the reception and transmission of plant hormone signals and related downstream signals. Combined with the results of previous studies, our work showed that the gene expression levels of *CLE9*, *FA*, *PUCHI*, *UF*, *CLV3*, *LOB30*, *SFT*, *S-WOX9* and *SVP* were significantly different in the two materials. The analyses of endogenous hormone and exogenous hormone induction showed that a variety of hormones were involved in tomato flowering reversion. These outcomes provide new insights into flower development in tomato and other plants and provide a basis for the study of optimal tomato plant types.

## Figures and Tables

**Figure 1 ijms-23-08992-f001:**
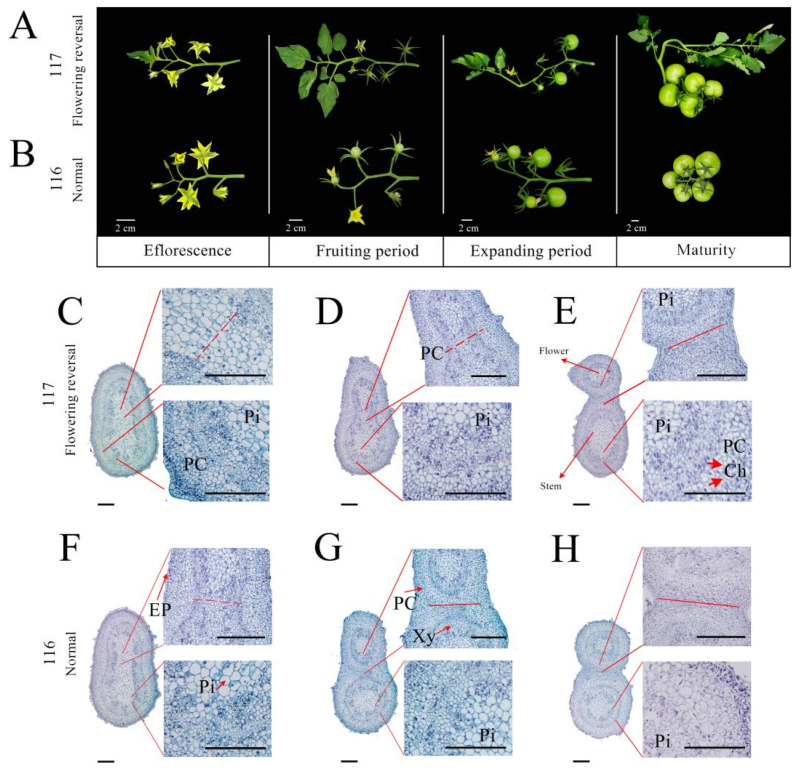
Phenotypic changes in the process of flowering reversion. (**A**,**B**) Phenotypes of flowering reversion with flower and fruit growth in ‘117’ (**A**) and ‘116’ (**B**) plants. (**C**–**H**) Microscopic observations of changes in flowering reversion in ‘117’ (**C**–**E**) and ‘116’ (**F**–**H**) at three different times (consistent with the Early, Mid and Late periods of the RNA-seq assay sampling time). The black solid line (**C**–**H**) represents a scale of 500 μm. The red line (**C**–**H**) represents the distinction between the flower Pi: Pith, PC: Parenchyma cell, Ch: Chlorophyll, EP: Epidermal, Xy: Xylem.

**Figure 2 ijms-23-08992-f002:**
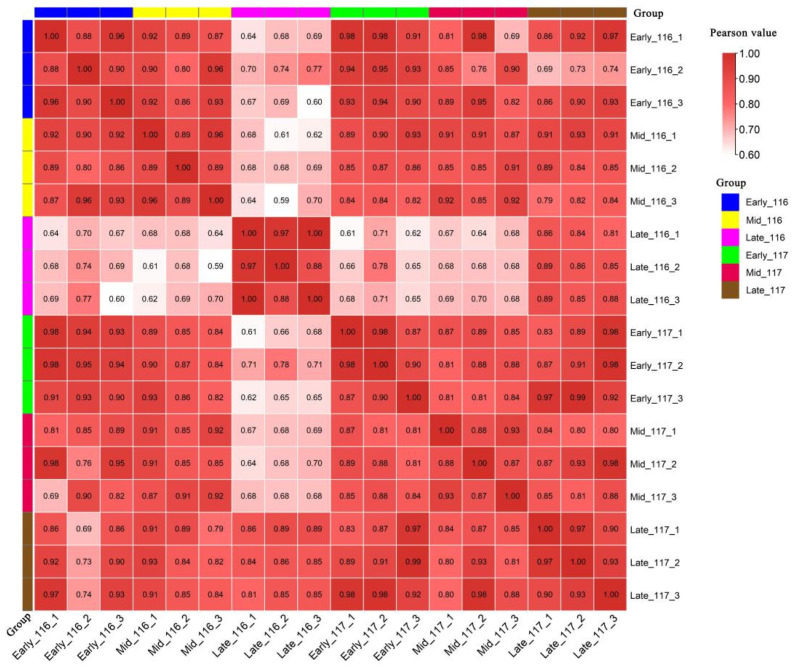
Individual correlation analyses between 18 transcriptome samples.

**Figure 3 ijms-23-08992-f003:**
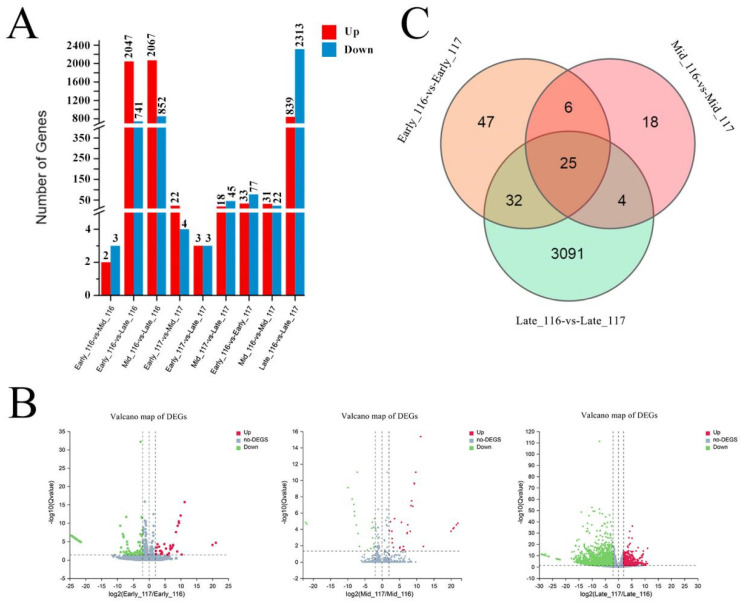
Statistics of the numbers of DEGs. (**A**) Statistical analysis of DEGs in different groups in different periods. (**B**) Volcano plot of DEGs between different groups. (**C**) Venn diagram statistics of comparisons between different groups.

**Figure 4 ijms-23-08992-f004:**
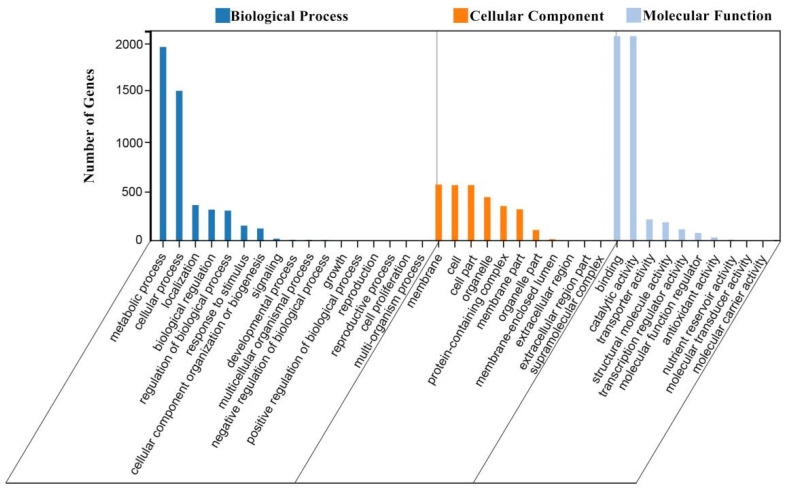
GO secondary classification of the set of DEGs in different periods between ‘116’ and ‘117’.

**Figure 5 ijms-23-08992-f005:**
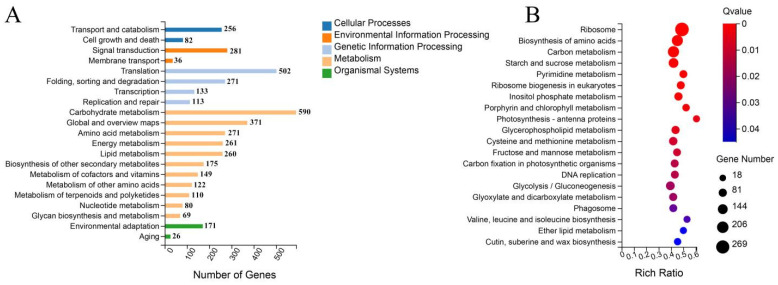
KEGG pathway enrichment analysis of the set of DEGs in different periods between ‘116’ and ‘117’. (**A**) Secondary classification of the KEGG pathways of DEGs. (**B**) Scatter plot of DEG enrichment in the top 20 KEGG pathways.

**Figure 6 ijms-23-08992-f006:**
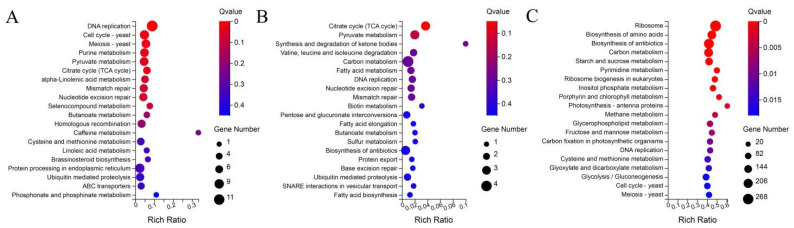
KEGG enrichment analysis of individual DEGs in three different periods. (**A**) KEGG analysis of DEGs at the early time point. (**B**) KEGG analysis of DEGs at the middle time point. (**C**) KEGG analysis of DEGs at the late time point.

**Figure 7 ijms-23-08992-f007:**
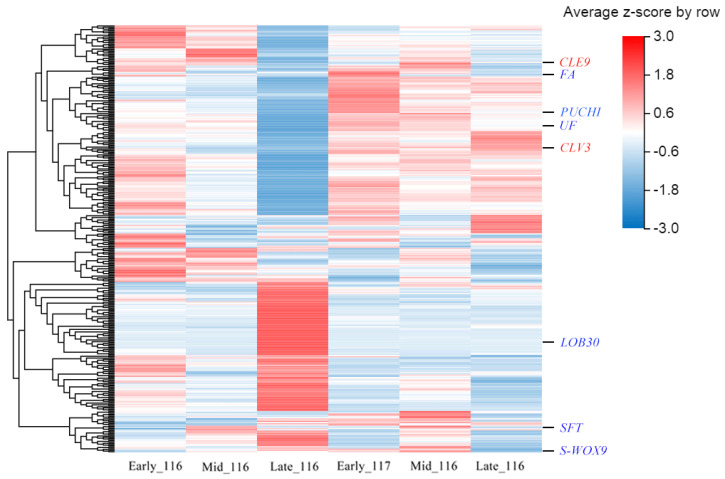
Heatmap of 351 gene clusters associated with flowering time.

**Figure 8 ijms-23-08992-f008:**
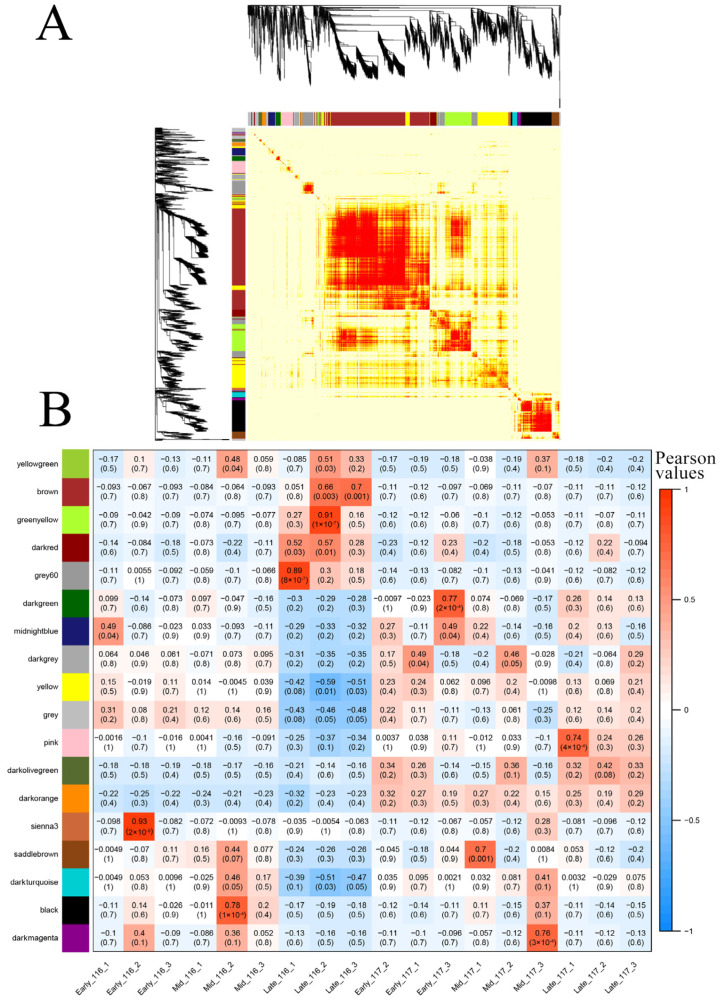
Gene co-expression network analysis by WGCNA. (**A**) Gene dendrogram colored according to the correlations between gene expression levels. Different colors represent different gene modules and indicate coefficients of dissimilarity between genes. (**B**) Module–sample association. The abscissa represents the samples; the ordinate represents the modules. The numbers in each cell are the correlation coefficients (**top**) and *p*-values (**bottom**).

**Figure 9 ijms-23-08992-f009:**
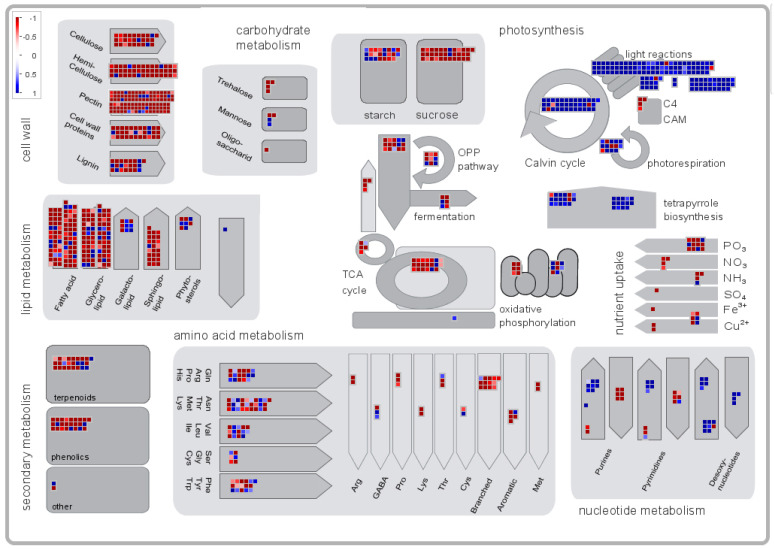
Regulatory overview produced by MapMan software.

**Figure 10 ijms-23-08992-f010:**
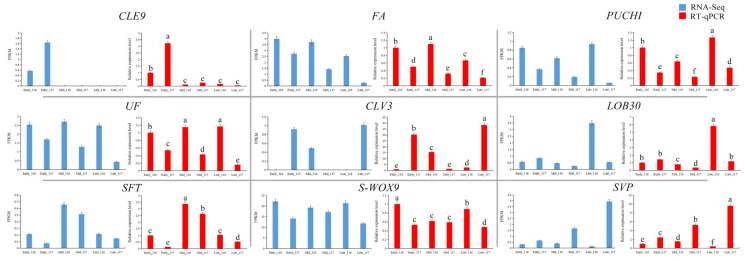
Comparative analysis of expression results between RNA-seq and RT-qPCR. Each bar represents the mean ± SD and different letters indicate significant differences (*t*-test, *p* ≤ 0.05).

**Figure 11 ijms-23-08992-f011:**
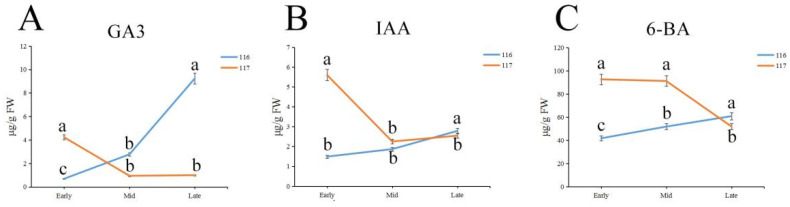
The levels of GA3 (**A**), IAA (**B**) and 6-BA (**C**) in inflorescence meristems of ‘116’ and ‘117’ plants. Each bar represents the mean ± SD and different lowercase letters indicate significant differences (*t*-test, *p* ≤ 0.05).

**Figure 12 ijms-23-08992-f012:**
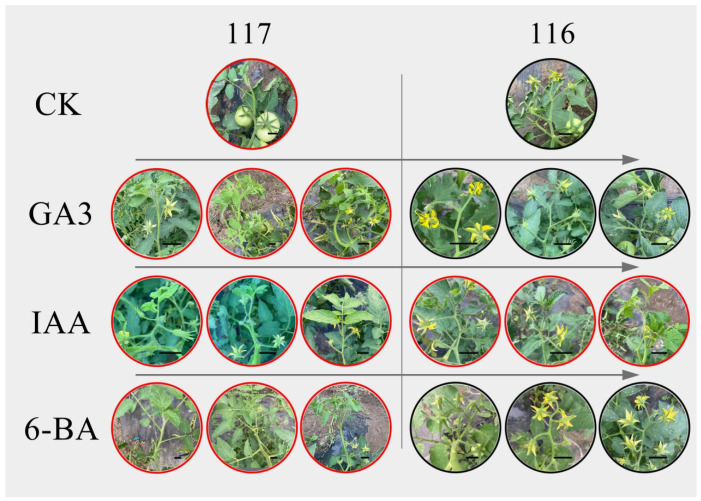
Plant phenotypes after hormone treatment. The red circles indicate the development of flowering reversion; the arrows indicate the gradual concentration increases. The black solid lines represent a scale of 2 cm.

## Data Availability

The raw sequencing data used in this article are stored in the NCBI Sequence Read Archive under accession number SUB10254346.

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
