# Peer review of "Transcriptome Analysis to Identify Genes Related to Flowering Reversion in Tomato"

_ijms, 2022, doi:10.3390/ijms23168992_

Round 1

Reviewer 1 Report

The manuscript is valuable and interesting. It is well written and organized. The topic fits within the scope of the journal. I only have a few minor suggestions that should be considered before publishing the text as listed below.

·        Keywords should be arranged alphabetically moreover, do not repeat words from the title.

·        Please provide full botanical names of species when mentioned for the first time, e.g., Solanum lycopersicum L.

·        Some of the photos lack scale bars.

·        Name of producer, city, state, and country should be given with key chemicals and equipment used, e.g. line 460.

·        Some parts of the Results are, in fact, a repetition of the Materials and methods section, e.g.l, lines 122-125.

·        Unit style is incorrect. Change mg/L to mg·L-1 everywhere.

·        The number of replications and repetitions is unclear.

·        Please, follow the MDPI formatting style in the Reference list.

After incorporating all the necessary changes, the manuscript can be accepted for publication.

Reviewer 2 Report

I am pleased to review the work "Transcriptome Analysis to Identify Genes Related to Flowering Reversion in Tomato". The manuscritp is generally written well. The introduction and discussion are very extensive. The aesthetically made figures also deserve attention. I believe that the work could be published in IJMS, although I have some comments and questions first.

1)    It is known that a lot of information can be obtained from the analysis of transcriptomes, as can be seen in this work. These are general data as well as very detailed data. In this case, is it possible to extract one or several genes that stand out the most ? For example, those highly or low expressed in a given system? Is it possible to attempt a broader analysis of its role or relationship?

2)   What is innovative at work, except that this type of analysis was performed for the first time on a tomato?

3)    It would be useful to mention specific genes that are crucial for the analysis in the Abstract.

4)  Keywords - we write all of them either in capital letters or in lower case.

5)    Materials and methods - lines 467-470 - please specify which kits were used for mRNA isolation, sequencing libraries and the sequencing itself. Please also write how many libraries were constructed and how many biological variants and repeats they represented. An experimental schema would be useful here.

6) Materials and methods - I can see the description of the analysis and statistical software used for RNASeq. However, there is no information about the tests and software used for other analyzes, for example, RT-qPCR.

7)   In my opinion, the correct name of the method that was used to validate the sequencing results is RT-qPCR and not qRT-PCR.

8)    Figure 10 - A comparison of the RNASeq and RT-qPCR results in the graphs could be presented more clearly. So that you could quickly compare one graph to another. The plots for each gene could be stacked side-by-side and distinguished by color. Or simply present the values ​​side by side in a table instead of graphs.

9)    Figure 11 - no description of individual letters - I suspect they refer to statistical analyzes.

10) Line 473 - after (1) - space

Author Response

Response to Reviewer 2 Comments

Dear Reviewer 2,

Thank you for your attention to our manuscript and recognition of our research content. We have carefully revised the manuscript according to your comments and suggestions. These modifications make our research content easier for readers to understand. Spell check and language fluency revision make our manuscript readable and rigorous. All changes in the manuscript have been highlighted. Point-by-point responses to comments are as follows:

Point 1: It is known that a lot of information can be obtained from the analysis of transcriptomes, as can be seen in this work. These are general data as well as very detailed data. In this case, is it possible to extract one or several genes that stand out the most ? For example, those highly or low expressed in a given system? Is it possible to attempt a broader analysis of its role or relationship?

Response 1: Thank you for your constructive comments.

As you mentioned, transcriptome is the link between genomic genetic information and biological function, and transcriptional regulation is the most important and widely studied way of biological regulation. Transcriptome research can provide more efficient and useful information than genome research. After finishing the work in this manuscript, we immediately carried out the functional analysis of the excavated key genes, such as CLV3, CLE9 and SVP genes. We are currently trying to construct a gene editing system, and yeast hybridization experiments are also being carried out simultaneously. More importantly, we are trying to locate the region of flowering reversion genes by GWAS and QTL methods. Finally, combined with transcriptome analysis, our study will be more attractive. These follow-up work requires a lot of time and materials, and we expect to publish the existing research results in the journal, which is not only a test of our phased research results, but also an encouragement for our follow-up research. In the near future, we will focus on the function of one or several of the most prominent genes, comprehensively analyze the role of these genes in tomato flowering reversion and write a research manuscript. We believe that the publication of the research manuscript will provide more reliable data support for our future research.

Point 2: What is innovative at work, except that this type of analysis was performed for the first time on a tomato?

Response 2: Thank you for your attention to the manuscript

In my opinion, the innovations of the research manuscript are as follows:

  • Phenotypic and microscopic observation of flowering reversion in tomato;
  • 3223 DEGs were identified by transcriptome analysis;
  • DEGs are involved in multiple biological processes;
  • Several plant hormones involved in flowering reversal in tomato.

Point 3: It would be useful to mention specific genes that are crucial for the analysis in the Abstract.

Response 3: Thank you very much for your kind advice.

Line 20-22 - As you mentioned, we have added this content to the Abstract.

Point 4: Keywords - we write all of them either in capital letters or in lower case.

Response 4: Thank you very much for your responsible comments.

Line 27,28 - The keywords part has been modified correctly.

Point 5: Materials and methods - lines 467-470 - please specify which kits were used for mRNA isolation, sequencing libraries and the sequencing itself. Please also write how many libraries were constructed and how many biological variants and repeats they represented. An experimental schema would be useful here.

Response 5: Thank you for your kind advice.

Line 487-493 - This part has been fully supplemented.

Point 6: Materials and methods - I can see the description of the analysis and statistical software used for RNASeq. However, there is no information about the tests and software used for other analyzes, for example, RT-qPCR.

Response 6: Thank you for your careful correction.

Line 544,545 - In section 4.9, we have the analysis and statistical software for RT-qPCR, and we have added PCR program settings.

Point 7: In my opinion, the correct name of the method that was used to validate the sequencing results is RT-qPCR and not qRT-PCR.

Response 7: Thank you for your careful review.

Line 300,302,305,313 - The point has been corrected correctly and the text has been carefully reviewed.

Point 8: Figure 10 - A comparison of the RNASeq and RT-qPCR results in the graphs could be presented more clearly. So that you could quickly compare one graph to another. The plots for each gene could be stacked side-by-side and distinguished by color. Or simply present the values side by side in a table instead of graphs.

Response 8: Thank you for your constructive comments.

Figure 10 has been optimized to make the results look more convenient and concise.

Point 9: Figure 11 - no description of individual letters - I suspect they refer to statistical analyzes.

Response 9: We are very sorry for the trouble caused to you.

Line 324-326 - We have optimized Figure 11 and explained the letters in the legend.

Point 10: Line 473 - after (1) - space

Response 10: Thank you for your careful correction.

Line 495 - This point has been corrected.

Finally, If there is any shortage, please inform us! We will try our best to revise this manuscript.

Thank you very much for your attention and kind advice.

Round 2

Reviewer 2 Report

Thank you very much. The authors responded to all my questions and comments and made the necessary corrections. I believe the work is worth publishing in IJMS. I wish the authors success in continuing this research.